# Increased Expression and Activation of FAK in Small-Cell Lung Cancer Compared to Non-Small-Cell Lung Cancer

**DOI:** 10.3390/cancers11101526

**Published:** 2019-10-10

**Authors:** Frank Aboubakar Nana, Delphine Hoton, Jérôme Ambroise, Marylène Lecocq, Marie Vanderputten, Yves Sibille, Bart Vanaudenaerde, Charles Pilette, Caroline Bouzin, Sebahat Ocak

**Affiliations:** 1Pole of Pneumology, ENT, and Dermatology (PNEU), Institut de Recherche Expérimentale et Clinique (IREC), Université catholique de Louvain (UCLouvain), 1200 Brussels, Belgium; frank.aboubakar@uclouvain.be (F.A.N.); marylene.lecocq@uclouvain.be (M.L.); marie.vanderputten@uclouvain.be (M.V.); yves.sibille@uclouvain.be (Y.S.); charles.pilette@uclouvain.be (C.P.); 2Division of Pneumology, Cliniques Universitaires St-Luc, UCLouvain, 1200 Brussels, Belgium; 3Department of Pathology, Cliniques Universitaires Saint-Luc, UCLouvain, 1200 Brussels, Belgium; delphine.hoton@uclouvain.be; 4Centre de Technologies Moléculaires Appliquées, IREC, UCLouvain, 1200 Brussels, Belgium; jerome.ambroise@uclouvain.be; 5Division of Pneumology, CHU UCL Namur (Godinne Site), UCLouvain, 5530 Yvoir, Belgium; 6Lung Transplant Unit, Division of Respiratory Disease, Department of Clinical and Experimental Medicine, Katholieke Universiteit Leuven, 3000 Leuven, Belgium; bart.vanaudenaerde@med.kuleuven.be; 7Imaging Platform, IREC, UCLouvain, 1200 Brussels, Belgium; caroline.bouzin@uclouvain.be

**Keywords:** expression, FAK, lung cancer, small-cell lung cancer, non-small-cell lung cancer, multiplex immunofluorescence staining, phospho-FAK, prognosis, targeted therapy

## Abstract

Introduction: Focal adhesion kinase (FAK) plays a crucial role in cancer development and progression. FAK is overexpressed and/or activated and associated with poor prognosis in various malignancies. However, in lung cancer, activated FAK expression and its prognostic value are unknown. Methods: FAK and activated FAK (phospho-FAK Y397) expressions were analyzed by multiplex immunofluorescence staining in formalin-fixed paraffin-embedded tissues from 95 non-small-cell lung cancer (NSCLC) and 105 small-cell lung cancer (SCLC) patients, and 37 healthy donors. The FAK staining score was defined as the percentage (%) of FAK-stained tumor area multiplied by (×) FAK mean intensity and phospho-FAK staining score as the (% of phospho-FAK-stained area of low intensity × 1) + (% of phospho-FAK-stained area of medium intensity × 2) + (% of the phospho-FAK-stained area of high intensity × 3). FAK and phospho-FAK staining scores were compared between normal, NSCLC, and SCLC tissues. They were also tested for correlations with patient characteristics and clinical outcomes. Results: The median follow-up time after the first treatment was 42.5 months and 6.4 months for NSCLC and SCLC patients, respectively. FAK and phospho-FAK staining scores were significantly higher in lung cancer than in normal lung and significantly higher in SCLC compared to NSCLC tissues (*p* < 0.01). Moreover, the ratio between phospho-FAK and FAK staining scores was significantly higher in SCLC than in NSCLC tissues (*p* < 0.01). However, FAK and activated FAK expression in lung cancer did not correlate with recurrence-free and overall survival in NSCLC and SCLC patients. Conclusions: Total FAK and activated FAK expressions are significantly higher in lung cancer than in normal lung, and significantly higher in SCLC compared to NSCLC, but are not prognostic biomarkers in this study.

## 1. Introduction

Lung cancer is histologically divided into two main types: Non-small cell lung cancer (NSCLC) and small cell lung cancer (SCLC), representing 85% and 15% of the cases, respectively [1,2]. In recent years, oncogenic drivers with sensitivity to targeted therapies (e.g., tyrosine kinase inhibitors (TKIs) targeting epidermal growth factor receptor (EGFR) mutations, anaplastic lymphoma kinase (ALK) rearrangements, or other oncogenic abnormalities) have been discovered in NSCLC, leading to improvements in the outcome of oncogenic-driven NSCLC patients [3]. Immunotherapy with anti-programmed death-ligand 1 (PD-L1) immune checkpoint inhibitors (ICIs) has also significantly improved the five-year overall survival (OS) of metastatic NSCLC patients without oncogenic drivers from 6% to 15% [4,5]. Clinically, SCLC is the most aggressive type of lung cancer, characterized by a high growth rate and a tendency for early metastasis, with two-thirds of the patients diagnosed with extensive stage (ES) disease and a five-year OS as low as 5% [2]. Despite improved understanding of the molecular steps leading to SCLC development and progression these last years, there is still no effective targeted therapies in SCLC, as opposed to NSCLC. After four decades, the only modest improvement in OS of patients suffering from ES-SCLC has been shown recently in a trial combining atezolizumab, an anti-PD-L1 immune checkpoint inhibitor, with carboplatin and etoposide, chemotherapy agents [6].

Focal Adhesion Kinase (FAK) is a 125 kDa cytosolic non-receptor tyrosine kinase widely expressed in various cell types and tissues. It is localized to focal adhesions or contact points between the actin cytoskeleton and the extracellular matrix. Once activated by integrins, G protein-coupled receptor ligands, or growth factors and neuromediators, FAK is autophosphorylated at tyrosine 397 (Y397), then binds and activates downstream proteins such as Src, p130CAS, paxillin, and PI3KR2 [7,8,9], finally leading to cell adhesion, migration, invasion, survival, proliferation, angiogenesis, immune suppression, and regulation of DNA damage repair [8,9,10,11]. Because of these roles and its overexpression in many cancers, with the correlation to poor prognosis in some of them [12,13,14,15,16,17,18,19,20,21,22,23], FAK is believed to play a role in cancer development and progression. Small-molecule inhibitors targeting the FAK kinase domain (e.g., PF-573,228) have, therefore, been developed as potential anti-cancer targeted therapies. They decreased FAK phosphorylation at Y397 and led to antitumoral effects in various cancer types, including NSCLC and SCLC [24,25,26,27]. In preclinical and clinical studies, they induced cancer regression or stability in several cancers, including NSCLC [26,28,29,30,31].

*FAK* gene copy number gain has previously been reported in 50% of 46 SCLC tissues analyzed by array comparative genomic hybridization and validated by fluorescent in situ hybridization and quantitative real-time polymerase chain reaction [32]. FAK activation has also been observed in SCLC cell lines, and inhibition of FAK phosphorylation at Y397 with PF-573,228 decreased cell proliferation, survival, migration, and invasion in SCLC cell lines [25]. These results suggested that FAK is important in SCLC biology and that targeting its kinase domain may have a therapeutic potential in SCLC patients. Moreover, total FAK expression has been evaluated by immunohistochemistry (IHC) in tissue microarrays (TMAs) including SCLC tissues from 85 patients, revealing an expression of FAK in 92% of the tumors, scored low in only 13%, while moderate in 20%, and strong in 59% of the samples [33]. However, no correlation was found between total FAK expression and recurrence-free survival (RFS) or OS in these SCLC patients [33]. Nevertheless, total FAK expression does not necessarily indicate an activated FAK pathway, as opposed to phospho-FAK expression.

Because there is a lack of data evaluating the expression of phospho-FAK in human lung cancer tissues as opposed to total FAK expression [19,33,34], we aimed to evaluate the expression of phospho-FAK (Y397) in SCLC and NSCLC tissues, and correlate the data to patients’ prognosis.

## 2. Materials and Methods

### 2.1. Patients and Tissues Samples

Formalin-fixed paraffin-embedded (FFPE) tissue blocks from patients with lung cancer and healthy donors were obtained from the tumor registry of Cliniques Universitaires St-Luc, CHU UCL Namur (Godinne Site), and Katholieke Universiteit Leuven. Lung cancer tissues were collected between January 2011 and February 2016 from 95 NSCLC and 105 SCLC patients at the time of diagnosis before any medical treatment. Normal lung samples, used as controls, were collected from 37 healthy donors between February 2016 and March 2019. All tumor sections were reviewed by an experienced lung cancer pathologist (D.H.), and only tumor sections with representative areas of tumor and adjacent lung parenchyma were included in the study. Sixty-seven of the NSCLC tissues were represented in TMAs (prepared in accordance with reported methods) [35,36], while none of the SCLC tissues were because they were all transbronchial or transthoracic biopsies, with no surgical specimens, as opposed to the NSCLC tissues.

Treatment was administered on an individual basis according to the disease stage and patient performance status as per the standard of care. All patients were followed with chart review until death or until data analysis of the manuscript. Clinical data were obtained from the tumor registry and hospital charts. Histological classification of the tumors was based on the World Health Organization criteria [37]. All tumors were staged according to the 7th lung cancer TNM pathological classification and staging system of the International Union Against Cancer (UICC) [38]. Patient characteristics are summarized in Table 1A,B. This study was approved by the institutional ethical review board (CHU UCL Namur (Godinne)) at each medical center (number of approval: 115/2014). The normal lung samples were obtained from unused lungs of donors and collected according to existing Belgian law and approved by the hospital’s ethical committee (S59648, S61653).

### 2.2. Multiplex Immunofluorescence Immunohistochemistry (mIF-IHC)

FFPE tissue blocks were sectioned at 5 µm. After deparaffinization in toluene and methanol, endogenous peroxidases were inhibited for 15 min in Bloxall (Vector Laboratories, Peterborough, UK) followed by 30 min in 0.3% hydrogen peroxide. Sections were then submitted to microwave antigen retrieval in 10 mM citrate pH 6.0 buffer containing 0.1% triton and to blocking of specific antigen-binding sites (Tris buffered saline (TBS) containing 5% normal goat serum and 0.1% Tween 20). The first primary antibody was incubated in TBS containing 1% normal goat serum and 0.1% Tween 20 and detected by corresponding horseradish peroxidase (HRP)-conjugated polymer secondary antibodies for 40 min at room temperature (RT). HRP was then visualized by tyramide signal amplification (TSA) using AlexaFluor-conjugated tyramides (Thermo Fisher Scientific, Paisley, UK). After a new citrate buffer incubation step, the same protocol was applied with other primary antibodies and different AlexaFluor or fluorescein-conjugated tyramides. In this study, 2 sequential incubations with phospho-FAK Y397 rabbit antibody (0.5 µg/mL, 1 h at RT; Thermo Fisher Scientific) and total FAK mouse antibody (5 µg/mL, overnight at 4 °C; Thermo Fisher Scientific) were performed. Total FAK and phospho-FAK Y397 were, respectively, revealed with TSA-conjugated fluorophores, AF647 and AF594 (1:150 dilution in 0.1 M Borate pH 7.8 buffer, 10 min at RT; Thermo Fisher Scientific). Finally, nuclei were counterstained with Hoechst 33342 (Thermo Fisher Scientific) diluted in TBS containing 10% BSA and 0.1% Tween 20, washed in TBS containing 0.1% Tween 20, and mounted with a Dako fluorescence mounting medium (Dako, Glostrup, Denmark). Negative controls were established by adding nonspecific isotype controls as primary antibodies. Slides were stored at −20 °C until multispectral image acquisition.

After the image acquisition, coverslips were removed by immersion of the slides into water overnight at room temperature (RT). Sections were incubated with 5% human serum before adding anti-pan cytokeratin CKAE1-AE3 antibody (1:200, 1 h at RT; Dako), followed by HRP-conjugated polymer secondary antibody for tumor detection. Peroxidase activity was revealed through 5 min incubation with diaminobenzidine (DAB) substrate (IM2394; Immunotech, Marseille, France). Slides were finally counterstained for 3 min with hematoxylin (Dako).

### 2.3. Stained Slides Imaging 

Multiplex immunofluorescence immunostained slides were digitalized in fluorescence using a Pannoramic 250 FlashIII scanner (3DHistech, Budapest, Hungary) at 20× magnification using the following filter cubes: DAPI1 (ex: 377/50 nm–em: 477/60 nm), SpRed (ex: 586/20 nm–em: 628/32 nm), and Cy5 (ex: 328/40 nm–em: 692/40 nm). After pan-CKAE1-AE3 staining, slides were re-scanned in brightfield at the same magnification.

### 2.4. Quantitative Evaluation of Immunostaining 

FAK and phospho-FAK stainings were quantified on multiplex-stained paraffin sections (TMA or not) with software applications (APP) using the image analysis tool Oncotopix version 2017.2 (Visiopharm, Hørsholm, Denmark). Using the first APP, the tumor was delineated based on the CKAE1-AE3 staining at low digital magnification (3×) using a thresholding classification method based on the HDAB-DAB feature of the software and post-processing steps. These are designed to fill the detected area and to outline it within a region of interest (ROI). For TMA sections, each TMA plug was outlined with a different ROI (Figure 1). Within the delineated tumor, CKAE1-AE3-stained tumor clusters were delineated in a second APP at low digital magnification (5×) using a thresholding classification method based on the HDAB-DAB feature of the software. Large empty spaces (alveoli, vessels, and damaged tissues) were discarded. Delineated scans were then duplicated to proceed in parallel to the detection and quantification of FAK and phospho-FAK at high magnification (20×).

FAK-stained areas were detected with a thresholding classification method on the Alexa fluor 647 staining. The mean fluorescence intensity of the pixels within the tumor cluster ROI was also calculated. FAK expression results were reported using a FAK staining score, corresponding to the percentage (%) of FAK-stained tumor area multiplied by (×) FAK mean intensity. Phospho-FAK stained areas were detected using 3 thresholds of intensity of the AlexaFluor594 to highlight the differences in staining intensity, much lower than the total FAK, and, therefore, requiring a different method of detection. Phospho-FAK expression results were reported using a phospho-FAK staining score, corresponding to (% of phospho-FAK-stained tumor area of low intensity × 1) + (% of phospho-FAK-stained tumor area of medium intensity × 2) + (% of phospho-FAK-stained tumor area of high intensity × 3). Similar calculations were used to evaluate FAK and phospho-FAK staining scores in the nuclei (detected with a thresholding classification method based on the Hoechst nuclear counterstaining).

### 2.5. Western Blot

Thirty frozen NSCLC, 10 frozen SCLC, and 9 frozen normal lung tissues were lysed with 250 µL of RadioImmunoPrecipitation Assay (RIPA) buffer with anti-protease and anti-phosphatase agents (Roche Diagnostics, Mannhein, Germany). Equal amounts of lysate were separated by 12% SDS-PAGE and electrotransferred onto a nitrocellulose membrane. After blocking 1 h with 5% W/V BSA (Sigma, Saint-Louis, MO, USA) in TBS with 0.1% Tween 20 (Sigma), the membrane was incubated overnight at 4 °C with phospho-FAK Y397 rabbit antibody (1/1000 Cell Signaling Technology, Danvers, MA, USA) or total FAK mouse antibody (1/250, Santa Cruz Biotechnology, Dallas, TX, USA) and glyceraldehyde 3-phosphate dehydrogenase (GAPDH) rabbit antibody (1/5000, Sigma). Secondary antibodies consisted of HRP-conjugated goat anti-rabbit IgG (Cell Signaling Technology) or HRP-conjugated goat anti-mouse IgG (Sigma). Immunoreactivity bands were developed using chemiluminescence (Amersham ECL, GE Healthcare, Little Chalfont, Buckinghamshire, UK) and detected with a chemidoc XRS apparatus (Bio-rad, Hercule, CA, USA) and quantified using the Quantity One software (Bio-rad).

### 2.6. Statistical Analysis

RFS and OS were computed for all patients as the time between first treatment (i.e., surgery, chemoradiation, or first-line chemotherapy) and the first relapse or death. Patients were right censored at the time of their last date of physical examination when they were still alive and without relapse at the time of analysis. Univariate and multivariate hazard ratios were computed on RFS using univariate and multivariate Cox proportional hazard regression models. *p*-values were obtained using linear models and adjusted for multiple testing using the Bonferroni method. All data were analyzed using R.3.4.0. A *p*-value *p* < 0.05 was considered to be statistically significant.

## 3. Results

### 3.1. Patient Characteristics

A total of 95 patients diagnosed with NSCLC were included in the study based on the availability of archival pathology specimens. NSCLC patient characteristics are described in Table 1A. Of the 95 patients, 26 (27.4%) were women, and 69 (72.6%) were men. Only four (4.2%) patients had never smoked, and one (1.1%) had an unknown smoking history, while all the others were current (*n* = 33, 34.7%) or ex-smokers (*n* = 57, 60.0%) with a median pack-year history of 40 (range: 2 to 107). The median age at diagnosis was 66 years (range: 30 to 85), with 21 (22.1%) patients older than 75 years. The disease was stage I for 45 (47.4%), II for 23 (24.2%), III for 19 (20.0%), and IV for 8 (8.4%) patients. All patients received medical treatment according to the disease stage and performance status as per the standard of care therapy. Recurrence free survival (RFS) was assessed according to Response Evaluation Criteria in Solid Tumors guidelines and was available for all patients. The median time of follow-up after the first treatment was 42.5 months (range: 1.3 to 92.4). At two and five years, the RFS was 76.6% (95% CI: 68.3–85.9%) and 67.5% (95% CI: 58.2–78.3%), respectively. At two and five years, OS was 80.9% (95% CI: 73.3–89.2%) and 66.2% (95% CI: 56.9–77.0%), respectively.

A total of 105 patients diagnosed with SCLC were included in the study based on the availability of archival pathology specimens. SCLC patient characteristics are described in Table 1B. Of the 105 patients, 38 (36.2%) were women, and 67 (63.8.6%) were men. Only one (1.0%) patient had never smoked, and 10 (9.5%) had an unknown smoking history, while all the others were current or ex-smokers with a median pack-year history of 43 (range: 1 to 170). The median age at diagnosis was 66 years (range: 43 to 89), with 27 (25.7%) patients older than 75 years. The disease stage was extensive for 69 (65.7%), limited for 21 (20.0%), and unknown for 15 (14.3%) patients. The median time of follow-up after the first treatment was 6.4 months (range: 0.1 to 79.0). At two and five years, the RFS was 13.1% (95% CI: 7.8–22.0%) and 7.6% (95% CI: 2.7–15.5%), respectively. At two and five years, the OS was 20.2% (95% CI: 13.7–29.7%) and 7.0% (95% CI: 3.3–15.0%), respectively.

A total of 37 healthy donors provided normal lung tissues. Their clinical characteristics are described in Table 1C. Of the 37 patients, 3 (8.1%) were women, and 34 (91.9%) were men. Eight of them (21.6%) had never smoked, 10 (27.0%) were current or ex-smokers, and 19 (51.4%) had an unknown smoking history. The median age at diagnosis was 59.5 years (range: 19 to 79).

### 3.2. FAK Expression and Activity Are Higher in SCLC than NSCLC and Normal Lung

FAK and phospho-FAK (Y397) staining pattern in NSCLC whole slide samples was homogenous, with either all the cells staining for FAK or phospho-FAK or none at all, although the staining intensity was clearly different between samples (Figure 2A–D). Based on this observation, it is concluded that the FAK staining quantification could also be performed on NSCLC samples organized in TMAs (available for NSCLC but not for SCLC).

FAK expression was mainly cytoplasmic, while phospho-FAK (Y397) staining was mainly nuclear, both in NSCLC (Figure 2A–D) and SCLC (Figure 2E). FAK staining was found in tumor and normal broncho-epithelial cells, as well as in immune and endothelial cells from the tumor microenvironment. Phospho-FAK was mainly expressed in tumor, endothelial, and some immune cells but not in the normal broncho-epithelial cells (Figure 2A–G). Of note, peritumoral normal lung and tumor microenvironment were observed only in the NSCLC samples because the SCLC biopsies were small and almost exclusively consisting of tumor cells.

FAK expression was significantly higher in SCLC compared with NSCLC and normal lung tissues as assessed by mean FAK staining scores (11863 ± 5798 vs. 8727 ± 4501 vs. 418 ± 468, respectively) (*p* < 0.01) (Figure 3A). FAK activity, represented by phospho-FAK (Y397) expression, was predominantly found in tumor cells, whereas its expression was low in normal lung alveoli and interstitial tissue (Figure 2A–D). Phospho-FAK (Y397) expression was significantly increased in SCLC compared with NSCLC and normal tissues as assessed by mean phospho-FAK staining scores (146 ± 50 vs. 67 ± 32 vs. 17 ± 11, respectively) (*p* < 0.01) (Figure 3B). Interestingly, the proportion of activated FAK compared with total FAK expression was significantly increased in SCLC as compared with NSCLC, as assessed by the ratio between mean phospho-FAK staining score and mean FAK staining score (0.025 ± 0.063 vs. 0.011 ± 0.014) (*p* < 0.01) (Figure 3C). 

In a second step, the FAK and phosphor-FAK nuclear staining were specifically evaluated. It was found that the mean nuclear FAK staining scores were significantly increased in lung cancer as compared to normal lung tissues, but without significant difference between NSCLC and SCLC (146.5 ± 61.4 vs. 130.4 ± 39.4 vs. 55.2 ± 19.5, respectively) (*p* < 0.01 only for comparison between normal and lung cancer) (Figure 4A), while mean nuclear phospho-FAK staining scores were significantly increased in SCLC as compared to NSCLC and normal lung samples (91 ± 47 vs. 37 ± 11 vs. 25 ± 12, respectively) (*p* < 0.01) (Figure 4B).

In order to validate the observations made by mIF-IHC, the FAK and phospho-FAK expression by western blot (WB) in 10 SCLC, 30 NSCLC, and nine normal lung tissue lysates were evaluated. This technique confirmed a significant increase in SCLC, compared with NSCLC and normal lung tissues, of FAK expression (0.177 ± 0.169 vs. 0.052 ± 0.066 vs. 0.013 ± 0.024, respectively) (*p* = 0.04) (Figure 5A–C) and phospho-FAK expression (0.727 ± 0.448 vs. 0.021 ± 0.053 vs. 0.056 ± 0.09) (*p* < 0.001) (Figure 5B,C).

### 3.3. FAK Expression and Activity Do Not Correlate with Patient Characteristics or Survival

The availability of the clinical data for each sample from the NSCLC (*n* = 95) and the SCLC (*n* = 105) patients enabled the assessment of the impact of FAK expression and activity on survival outcomes. Univariate analysis of staining scores treated as continuous variables showed no significant correlation of FAK and phospho-FAK expression with RFS and OS in NSCLC (Table 2A) and SCLC (Table 2B) patients.

In a multivariate analysis including disease stage, age at diagnosis, smoking history, and histological subtype, no significant association was found between phospho-FAK staining score and RFS or OS in NSCLC patients (Table 3A). Similarly, in a multivariate analysis including disease stage, age at diagnosis, and smoking history, the ratio between phospho-FAK staining score and FAK staining score was not significantly associated with RFS or OS in SCLC patients (Table 3B). As expected, the disease stage was the most significant independent predictor of RFS and OS in both NSCLC and SCLC. 

## 4. Discussion

In this study, with mIF-IHC, it is shown that FAK and phospho-FAK are both significantly overexpressed in lung cancer as compared to normal lung tissues. Interestingly, we also showed that, among lung cancers, FAK and phospho-FAK expression, as well as the ratio between phospho-FAK and FAK expression, are significantly higher in SCLC compared to NSCLC. Moreover, these observations were validated by WB of NSCLC, SCLC, and normal lung tissue lysates. However, no correlation was found between FAK and activated FAK expression in lung cancer and RFS or OS in NSCLC and SCLC patients.

The overexpression and activation of FAK that was observed in lung cancer tissues from treatment-naïve patients, as compared to normal lung tissues, was compatible with the well-known role of FAK in cancer initiation and progression. Indeed, FAK is known to promote cell proliferation, survival, migration, invasion, angiogenesis, and immune suppression in several cancers, including lung cancer [9,11,25,31,39,40,41]. 

As in our study, FAK overexpression has previously been reported in many cancers [42], including NSCLC and SCLC [19,33,34,43,44,45,46]. Moreover, FAK overexpression has been associated with poor survival in various cancers [12,14,20,22,47,48]. In NSCLC, however, discordant results have been reported. In two studies, including 381 [46] and 249 [44] patients with stage I-III NSCLC, FAK overexpression evaluated by IHC has been correlated with poor OS. Additionally, FAK overexpression evaluated by IHC has been correlated with increased lymph node metastasis, more advanced disease stages, and poor prognosis in a study of 153 patients with stage I–III NSCLC [19]. Nevertheless, similarly to our study, the prognostic value of FAK overexpression has not been found in a cohort of 103 patients with stage I NSCLC [34]. In SCLC, FAK expression has not been associated either with RFS and OS [33].

Unlike total FAK expression, phospho-FAK (Y397) expression represents the activation status of FAK [11,49,50] and is, therefore, expected to be a more relevant biomarker. Aggregation of FAK with integrins and cytoskeletal proteins in focal adhesion contacts is the best described mechanism leading to FAK activation through phosphorylation of Y397. However, it is also well known that FAK can be activated by extracellular growth factors, including those released by lung cancer, such as bombesin, gastrin-related peptide (GRP), HGF, VEGF, TGF-β, HGF, and FGF [51,52,53,54,55,56,57,58,59]. This relationship between FAK and growth factors and neuroendocrine mediators could underline the preferential activation of FAK observed in this study in SCLC tissues, as compared to NSCLC. Increased FAK activity has already been reported in various cancer cell lines, with demonstrated antitumoral effects of FAK TKI in cancer cells where FAK was activated [27,30,40,60,61,62,63], including NSCLC and SCLC [24,25,26,39]. In a recent study, FAK inhibition with PF-573,228, a small-molecule TKI, decreased proliferation, survival, migration, and invasion in SCLC cell lines [25,32]. Similar results have also been demonstrated in NSCLC cell lines, where FAK TKI decreased cell viability [24,39]. However, data related to FAK activation status in human cancer samples are scarcer. In a study including 59 patients with stage I–IV gastric carcinomas, phospho-FAK (Y397) expression evaluated by IHC was correlated with poor five-year RFS after surgery. Interestingly, multivariate analyses showed that phospho-FAK was an independent predictor of gastric cancer recurrence rather than total FAK expression [64]. In a study of 113 patients with stage II osteosarcoma, high FAK and phospho-FAK expression by IHC were associated with poor metastasis-free and OS [65]. This result was consistent with the prognostic and predictive value of phospho-FAK overexpression reported in another study of 53 metastatic osteosarcomas [48]. In NSCLC, expression of phospho-FAK has been evaluated by WB in 44 stage I–III NSCLC frozen tissues, revealing an increased expression in NSCLC compared with normal lung tissues [66]. Furthermore, in the same study, increased phospho-FAK expression was correlated with higher nodal involvement of cancer and a poorer RFS [66]. In another study where phospho-FAK (Y397) expression was evaluated by IHC in 145 NSCLC tissues, overexpression was found but not correlated with survival [67]. 

To the best of our knowledge, FAK activity has not been previously reported in SCLC human tissues. Our study is, therefore, the first report of phospho-FAK expression in SCLC. It is also the first report of FAK and phospho-FAK expression in a large cohort of both SCLC and NSCLC. Moreover, we provide the first comparison of FAK expression and activation status between NSCLC and SCLC tissues, showing that total FAK expression and FAK activity are both significantly higher in SCLC than in NSCLC, which suggests that the FAK pathway is more activated in SCLC than in NSCLC. Based on this observation, it is also hypothesized that the higher activation of FAK in SCLC than in NSCLC is responsible for the more aggressive biological and clinical behavior of SCLC, known for the rapid growth, early, and frequent metastasis, and the poorest OS among all lung cancer types. Finally, high FAK activity in SCLC suggest that FAK may be a good anti-cancer target in SCLC, alone or in combination with chemotherapy, immunotherapy, and/or radiotherapy.

Despite the lack of prognostic value of total FAK and phospho-FAK expression in SCLC and NSCLC, a predictive value is not to be ruled out. Several FAK TKI have been tested in clinical trials, including patients suffering from various advanced-stage cancers, which showed their antitumoral activity (up to 33% objective response rates) and safety [30,63,68]. However, there is still no biomarker to identify patients likely to respond to FAK TKI. Thus, our findings provide a framework for clinical trials evaluating FAK TKI to test, prospectively, the total FAK and activated FAK expression, as well as the ratio between the activated FAK and total FAK as potential predictive biomarkers of response to FAK TKI. This would be especially relevant for SCLC patients, facing limited and disappointing therapeutic options, with the absence of effective targeted therapies. 

In the meantime, it would also be interesting to prospectively correlate FAK expression and activity in formalin-fixed paraffin-embedded (FPPE) human lung cancer tissues with response rates to FAK TKI of corresponding patient-derived xenograft models (immediate transfer of human cancer cells from NSCLC or SCLC patients to recipient immunodeficient mice).

The innovative mIF-IHC staining and quantification method used provides an accurate FAK and phospho-FAK expression evaluation in lung cancer. This accuracy allowed us to specifically analyze FAK and phospho-FAK expression in the nucleus. Besides the well-known role of FAK in the cytosol downstream of integrin and growth factor receptor signaling, it has been shown that FAK also plays a functional role in the nucleus, where it can enter, bind to transcription factors, and regulate gene expression to influence tumorigenesis [50,69,70,71]. In this study, it is shown for the first time that nuclear FAK and phospho-FAK expression is significantly higher in SCLC than in NSCLC and normal lung. Furthermore, this accuracy of mIF-IHC would be particularly relevant to evaluate and quantify FAK expression and activation status in tumor microenvironment where FAK has been shown to play a crucial role in antitumor immune evasion, for instance in pancreatic cancer [40,72]. Finally, the mIF-IHC method requires smaller amounts of sample than conventional IHC and is, therefore, valuable when limited tumor tissue is available, as it is usually the case in SCLC where surgical specimens are scarce because patients are rarely treated by surgery.

## 5. Conclusions

Analysis of 105 SCLC, 95 NSCLC, and 37 normal lung tissues revealed that FAK expression and activity are both significantly higher in SCLC compared with NSCLC and normal lung tissues. This suggests that the FAK pathway is more activated in SCLC than in NSCLC and that FAK may be a good anti-cancer target in SCLC, alone or in combination with chemotherapy, immunotherapy, and/or radiotherapy. Although our study did not find any correlation between FAK expression or activity and survival, suggesting that they are not prognostic biomarkers in lung cancer patients, the present workflow may be used to further assess FAK expression and activity as predictive biomarkers of response (theranostic biomarker) to FAK TKI in future clinical trials. 

## Figures and Tables

**Figure 1 cancers-11-01526-f001:**
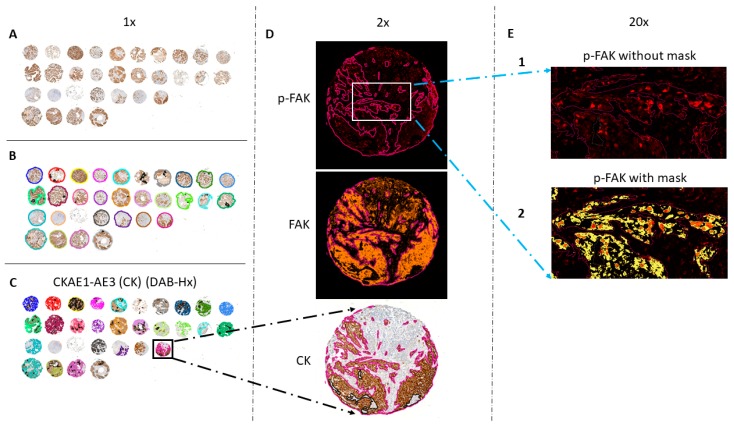
Illustration of focal adhesion kinase (FAK) and phospho-FAK staining quantification on a tissue microarray section of non-small-cell lung cancer (NSCLC) stained by multiplex immunofluorescence (IF) immunohistochemistry (IHC). (**A**) Tissue microarray (TMA) sections were sequentially stained by mIF with an antibody against phospho-FAK (red signal) and FAK (orange signal), followed by the Hoechst nuclear marker (blue signal). After whole slide fluorescence image acquisitions, IHC was performed with a tumor marker using an antibody against pan-cytokeratin CKAE1-AE3 (CK, brown signal) on the same slide and digitalized with a slide scanner. (**B**) Each TMA plug was then automatically delineated via the image analysis tool Oncotopix version 2017.2 (Visiopharm). (**C**) CK-positive tumor regions were semi-automatically delineated from CK-negative stroma. (**D**) These tumor regions, detected on the brightfield scan, were transposed to the aligned fluorescent scan with the Visiopharm Tissue Align module. (**E**) FAK and phospho-FAK stained areas were finally detected and quantified as illustrated for phospho-FAK in Figure D.2., with staining detection according to three thresholds of intensity (low, yellow; medium, orange; high, red), while Figure D.1. shows phospho-FAK staining without the mask. Original magnification: A, B, C: 1×; D: 2×; E: 20×.

**Figure 2 cancers-11-01526-f002:**
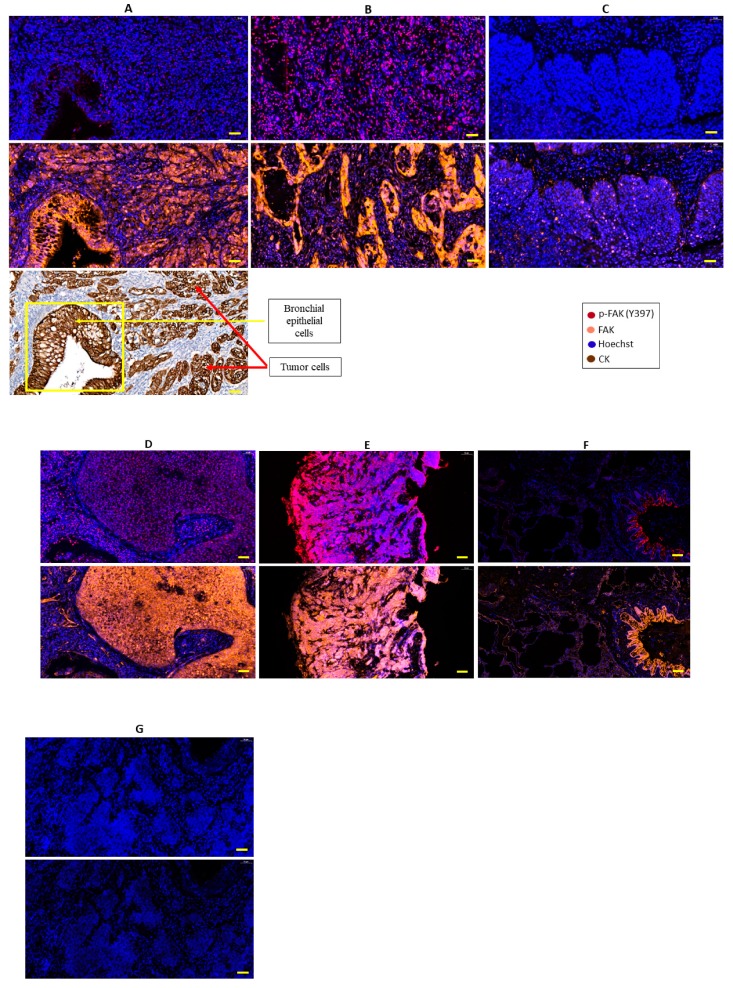
Illustrations of FAK and phospho-FAK (Y397) expression evaluated by multiplex immunofluorescence (IF) immunohistochemistry (IHC) in lung cancer and normal lung tissues. (**A**) Lung adenocarcinoma with the absence of phospho-FAK expression but homogenous cytoplasmic FAK staining (orange) in the tumor core, adjacent non-tumoral bronchi, and some stromal cells (including vessels and lymphoid structures). (**B**) Lung adenocarcinoma with nuclear phospho-FAK staining (red) and homogenous cytoplasmic FAK staining (orange). (**C**) Lung squamous carcinoma with the absence of phospho-FAK expression but weak cytoplasmic FAK staining. (**D**) Lung squamous carcinoma with nuclear phospho-FAK staining (red) and homogenous cytoplasmic FAK staining (orange). (**E**) Small-cell lung cancer with nuclear phospho-FAK staining (red) and cytoplasmic FAK staining (orange). (**F**) Normal lung with cytoplasmic FAK staining in bronchi and some stromal cells (including vessels and lymphoid structures). (**G**) Lung squamous carcinoma used as a negative control, showing the absence of phospho-FAK and FAK staining. Original magnification: 20×; scale bar: 50 µm.

**Figure 3 cancers-11-01526-f003:**
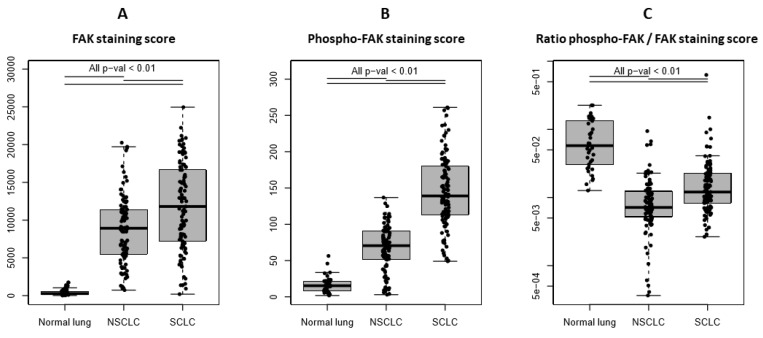
Quantification of FAK and phospho-FAK (Y397) expression evaluated by multiplex immunofluorescence immunohistochemistry in 37 normal lungs, 95 non-small-cell lung cancer (NSCLC), and 105 small-cell lung cancer (SCLC) tissues: (**A**) FAK staining score: Percentage (%) of FAK-stained tumor area multiplied by (x) FAK mean intensity, (**B**) phospho-FAK (Y397) staining score: (% of phospho-FAK-stained tumor area of low intensity × 1) + (% of phospho-FAK-stained tumor area of medium intensity × 2) + (% of phospho-FAK-stained tumor area of high intensity × 3), and (**C**) ratio between phospho-FAK and FAK staining scores. Each dot represents one sample. Data presented as the mean ± S.D. *p*-values were obtained using linear models and adjusted for multiple testing using the Bonferroni method.

**Figure 4 cancers-11-01526-f004:**
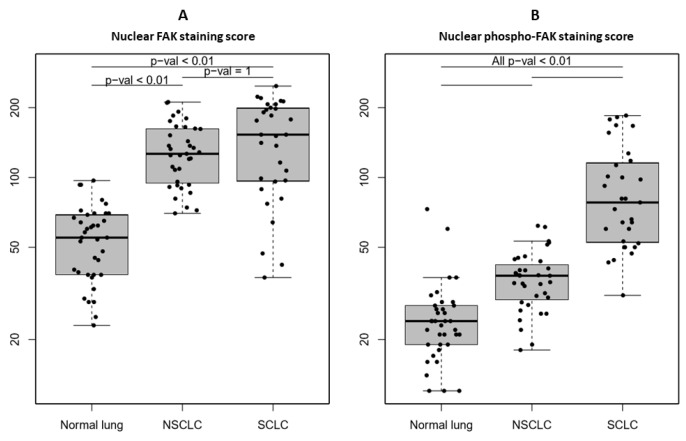
Quantification of nuclear FAK and nuclear phospho-FAK (Y397) expression evaluated by multiplex immunofluorescence immunohistochemistry in 37 normal lung, 95 non-small-cell lung cancer (NSCLC), and 105 small-cell lung cancer (SCLC) tissues: (**A**) Nuclear FAK staining score: Percentage (%) of FAK-stained nucleus area multiplied by (×) nuclear FAK mean intensity, (**B**) nuclear phospho-FAK (Y397) staining score: (% of phospho-FAK-stained nucleus area of low intensity × 1) + (% of phospho-FAK-stained nucleus area of medium intensity × 2) + (% of phospho-FAK-stained nucleus area of high intensity × 3). Each dot represents one sample. Data presented as the mean ± S.D. *p*-values were obtained using linear models and adjusted for multiple testing using the Bonferroni method.

**Figure 5 cancers-11-01526-f005:**
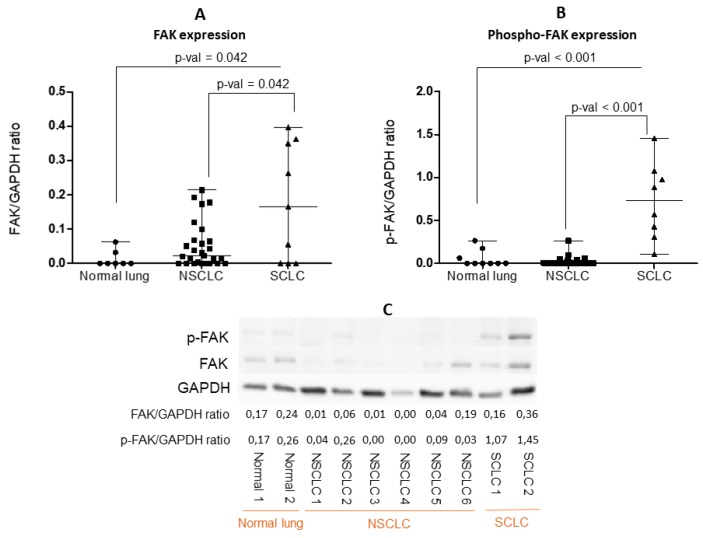
Quantification of (**A**) FAK and (**B**) phospho-FAK expression evaluated by Western blot (WB), with normalization to glyceraldehyde 3-phosphate dehydrogenase (GAPDH ) expression, in nine normal lungs, 30 non-small-cell lung cancer (NSCLC), and 10 small-cell lung cancer (SCLC) tissue lysates. Each dot represents one sample. Data presented as the mean ± S.D. Significance determined by the Kruskal-Wallis test. (**C**) Illustration of a representative WB of FAK and phospho-FAK (Y397) expression in normal lung, NSCLC, and SCLC tissue lysates. All the WB are represented in Appendix A.

**Table 1 cancers-11-01526-t001:** Clinical and pathological characteristics of (A) NSCLC patients, (B) SCLC patients, and (C) healthy donors.

A	
Variable	NSCLC Patients (*n* = 95)
Histological subtype	
Squamous	34 (35.8)
Non-squamous	61 (64.2)
Stage-*n* (%)	
I	45 (47.4)
II	23 (24.2)
III	19 (20.0)
IV	8 (8.4)
Gender-*n* (%)	
Female	26 (27.4)
Male	69 (72.6)
Age at diagnostic >75 y/o-*n* (%)	
No	74 (77.9)
Yes	21 (22.1)
Smoking history-*n* (%)	
Unknown	1 (1.1)
Never	4 (4.2)
Ex	57 (60.0)
Current	33 (34.7)
Pack per year	
Median	40
Range	2–107
B	
**Variable**	**SCLC Patients (*n* = 105)**
Stage-*n* (%)	
ED	69 (65.7)
LD	21 (20.0)
Unknown	15 (14.3)
Gender-*n* (%)	
Female	38 (36.2)
Male	67 (63.8)
Age at diagnostic >75 y/o-*n* (%)	
No	78 (74.3)
Yes	27 (25.7)
Smoking history-*n* (%)	
Unknown	10 (9.5)
Never	1 (1.0)
Ex	32 (30.5)
Current	62 (59.0)
Pack per year	
Median	43
Range	1–170
**C**	
**Variable**	**Healthy Donors (*n* = 37)**
Gender-*n* (%)	
Female	3 (8.1)
Male	34 (91.9)
Age >75 y/o-*n* (%)	
No	36 (97.3)
Yes	1 (2.7)
Agee (year)	
Median	59.5
Range	19−79
Smoking history-*n* (%)	
Unknown	19 (51.4)
Never	8 (21.6)
Yes	10 (27.0)

Abbreviations: ED, extensive-stage disease; LD, limited-stage disease; NSCLC, non-small-cell lung cancer; SCLC, small-cell lung cancer, y/o: Years old.

**Table 2 cancers-11-01526-t002:** Correlation of FAK and phospho-FAK expression with recurrence-free survival and overall survival in (A) NSCLC and (B) SCLC patients in a univariate analysis.

A				
Variable	Recurrence-Free Survival		Overall Survival	
	HR (95% CI)	*p*-Value	HR (95% CI)	*p*-Value
FAK staining score	1.00 (1.00–1.00)	0.92	1.00 (1.00–1.00)	0.99
Phospho-FAK staining score	0.99 (0.98–1.00)	0.1	0.99 (0.98–1.00)	0.21
Ratio phospho-FAK/FAK staining scores	0.86 (0.60–1.23)	0.33	0.89 (0.64–1.21)	0.4
B				
**Variable**	**Recurrence-Free Survival**		**Overall Survival**	
	**HR (95% CI)**	***p*** **-Value**	**HR (95% CI)**	***p*** **-Value**
FAK staining score	1.00 (1.00–1.00)	0.76	1.00 (1.00–1.00)	0.66
Phospho-FAK staining score	1.00 (1.00–1.01)	0.5	1.00 (0.99–1.00)	0.8
100 × ratio phospho-FAK/FAK staining scores	0.96 (0.90–1.02)	0.13	1.00 (0.97–1.03)	0.75

Abbreviations: CI, confidence interval; FAK, focal adhesion kinase; HR, hazard ratio.

**Table 3 cancers-11-01526-t003:** Multivariate Cox proportional regression analysis for the association with recurrence-free survival and overall survival of (**A**) phospho-FAK staining score in NSCLC (*n* = 95) and (**B**) the ratio between phospho-FAK and FAK staining scores in SCLC patients (*n* = 105).

A				
Variable	Recurrence-Free Survival		Overall Survival	
	HR (95% CI)	*p*-Value	HR (95% CI)	*p*-Value
Stage (ref: I)				
II	1.06 (0.35–3.20)	0.92	0.63 (0.19–2.04)	0.44
III	1.92 (0.72–5.16)	0.19	2.76 (1.11–6.83)	0.03
IV	5.64 (1.79–17.7)	<0.01	3.21 (1.06–9.66)	0.04
Age at diagnostic (ref: <75 y/o)				
>75 y/o	1.19 (0.49–3.07)	0.72	0.75 (0.28–2.03)	0.57
Smoking history (ref: ex and never)				
Current	0.82 (0.35–1.89)	0.64	0.88 (0.39–1.96)	0.75
Histology (ref: ADC)				
SCC	0.66 (0.26–1.69)	0.39	1.59 (0.70–3.62)	0.27
Phospho-FAK staining score	0.99 (0.98–1.00)	0.15	0.99 (0.98–1.01)	0.30
**B**				
**Variable**	**Recurrence-Free Survival**		**Overall Survival**	
	**HR (95% CI)**	***p*** **-Value**	**HR (95% CI)**	***p*** **-Value**
Stage (ref: LD)				
ED	3.75 (2.08–6.78)	<0.01	3.47 (1.91–6.32)	<0.01
Age at diagnostic (ref: <75 y/0)				
>75 y/o	1.56 (0.93–2.63)	0.09	1.42 (0.88–2.29)	0.15
Smoking history (ref: ex and never)				
Current	1.23 (0.78–1.93)	0.37	1.11 (0.73–1.70)	0.62
100 × ratio phospho-FAK/FAK staining scores	0.95 (0.89–1.02)	0.14	1.00 (0.97–1.03)	0.87

Abbreviations: ADC, adenocarcinoma; CI, confidence interval; ED, extensive-stage disease; LD, limited-stage disease; FAK, focal adhesion kinase; HR, hazard ratio; SCC, squamous cell carcinoma; y/0: Years old.

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
