# Peer review of "Increased Expression and Activation of FAK in Small-Cell Lung Cancer Compared to Non-Small-Cell Lung Cancer"

_cancers, 2019, doi:10.3390/cancers11101526_

Round 1

Reviewer 1 Report

The quality of the manuscript increased in the review process. The inclusion of several specimens from healthy controls further highlights the increased expression of FAK in tumor samples.

Analysis of pFAK in the nucleus is also novel and interesting for the field.

Figure legends are improved, as well as the discussion part.

Reviewer 2 Report

The authors have addressed all my concerns. 

This manuscript is a resubmission of an earlier submission. The following is a list of the peer review reports and author responses from that submission.

Round 1

Reviewer 1 Report

The authors studied the levels of phospho-focal adhesion kinase on small cell lung cancer tissue microarays to determine its relationship to survivability. The FAK, p-FAK and p-FAK/FAK levels in NSCLC were compared to those in SCLC. Comparisons should have been made with normal lung tissues for better understanding of the role of FAK and p-FAK in SCLC. Comparing disease tissues to other disease tissues and drawing conclusions on the relavance of the factors under investigation may lead to the wrong conclusions when comparisons to normal tissues are not also made. If this was done, perhaps an explanation for the 33% response of cases to FAK Kinase inhibitors would emerge. 

Reviewer 2 Report

In this study, the authors compared the difference in FAK and its Y397 phosphorylation between small cell lung cancer (SCLC) and non-small cell lung cancer using the IHC technique, and concluded that the expression levels of FAK and phospho-FAK (Y397) in SCLC were significantly higher than those in NSCLC, but were not correlated with patient characteristics or survival, and thus not considered a prognostic marker.

The authors did a fantastic job using the immunofluoresence double staining and IHC techniques to demonstrate the differential expression levels of FAK and phospho-FAK (Y397) in human SCLC and NSCLC tissue sections. However, it would be more convincing if authors could use a different technique, such as Western blotting, ELISA, real-time PCR, etc. to verify and confirm their IHC results. Otherwise, there could be a potential bias. Moreover, this study offered little insight into the possible mechanism underlying the relatively high expression of FAK in SCLC as compared with NSCLC. It would be more informative if more experiments could be included in the study to explore the molecular underpinning of the IHC observations. Otherwise, since the authors have concluded that FAK is not a prognostic marker, the significance of the study is limited as not further study would be needed. From what the authors have presented in the manuscript, it seems to me that FAK may not be the primary driver of the progression of SCLC and NSCLC, but may be activated and become a crucial bypass route when the primary oncogenic drivers (such as EGFR or K-Ras/B-Raf) are being blocked. 

Reviewer 3 Report

The paper by Nana et al, analyzed the expression of the activate form of FAK in its phosphorylated form Y397 in a vast cohort of patients with both NSCLC and SCLC cancer. In detail, they analyzed 105 and 95 patient samples by multiplex immunofluorescence immunochemistry to detect the expression of pFAK in primary tumor samples. Even if the expression of the pFAK was higher in SCLC than in NSCLC there was no correlation with OF and RFS.

Pro:

big cohort of primary tumor samples first analysis of pFAK in patient samples with SCLC

Contro:

only descriptive, only one marker analysed by one technique no healthy controls to compare

You find nuclear localization of FAK in both types of tumors. This finding is in line with other recent reports in the literature, supporting the idea of novel functions of FAK. It would be interesting to quantify the intensity of nuclear total and pFAK and to look if there are differences in the intensity of pFAK or total nuclear FAK between SCLC and NSCLC. It would be important also to add some CTRL samples, from healthy donors.

Lung cancer is a highly mutated tumor, with P53 and KRAS mutation ranking the highest. Do you have access to the mutation history of the patient? Would you be able to correlate the mutation background with pFAK expression?

Minor: 

Line 21: add space in thepercentage

Line 42: remove double space

Line 8:1 remove space

Figure legend 3: Add a description for the graph such as: each dot represent…Since you are comparing two groups of patients, I think it should be more appropriate to use Mann Whitney test.

Figure legend 2: too details, already explained.

Be more concise in the figure legends, add scale bar if possible on images and legends.

Discussion is too long in the first part, where you discuss previous data from others and poor in the part regarding your data.